# We Are What We Repeatedly Do:
# Inducing and Deploying Habitual Schemas in Persona-Based Responses

**Benjamin Kane**
University of Rochester
bkane2@ur.rochester.edu

**Lenhart Schubert**
University of Rochester
schubert@cs.rochester.edu

## Abstract

Many practical applications of dialogue technology require the generation of responses according to a particular developer-specified persona. While a variety of personas can be elicited from recent large language models, the opaqueness and unpredictability of these models make it desirable to be able to specify personas in an explicit form. In previous work, personas have typically been represented as sets of one-off pieces of self-knowledge that are retrieved by the dialogue system for use in generation. However, in realistic human conversations, personas are often revealed through story-like narratives that involve rich *habitual* knowledge – knowledge about kinds of events that an agent often participates in (e.g., work activities, hobbies, sporting activities, favorite entertainments, etc.), including typical goals, subevents, preconditions, and postconditions of those events. We capture such habitual knowledge using an explicit *schema* representation, and propose an approach to dialogue generation that retrieves relevant schemas to condition a large language model to generate persona-based responses. Furthermore, we demonstrate a method for bootstrapping the creation of such schemas by first generating *generic passages* from a set of simple facts, and then inducing schemas from the generated passages.

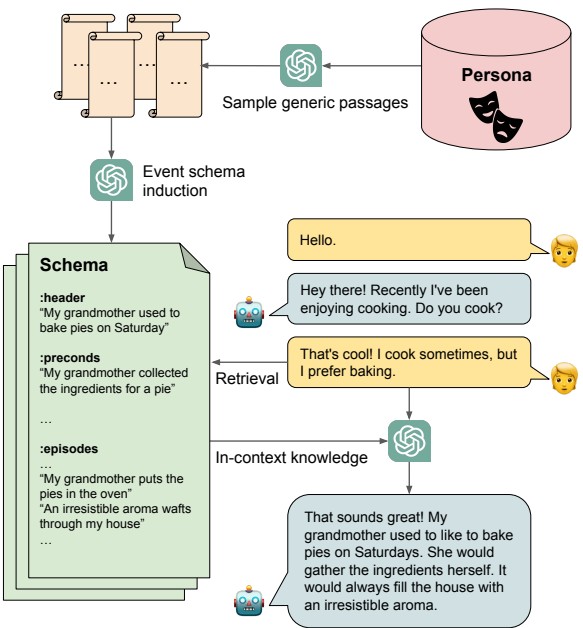

Figure 1: A diagram of our approach. (1) Given an unstructured persona dataset, we first sample "generic passages" from the facts in the persona, and then induce structured event schemas from the sampled stories. (2) We condition an LLM to generate dialogue responses that are fluent with previous conversation – yet that make use of the rich knowledge contained in the resulting schemas – by first using a retrieval model to select a relevant schema, and then providing the schema to the LLM as in-context knowledge.

## 1 Introduction

Virtual conversational agents – simulated humans that can engage in conversation with a human user – present a major application of dialogue technology. Such systems have been deployed for diverse uses including conversational coaches, chatbots for entertainment, and customer service bots. A critical, yet challenging, problem in designing conversational agents is endowing them with a specific *persona*, and generating responses that are both natural and consistent with this persona. Systems that are able to do this are both found to be more engaging by users (Zhang et al., 2018), and increase

the level of confidence and trust that users place in the system (Shum et al., 2018). Furthermore, in many practical applications beyond chit-chat, there is a complementary need to control the flow of dialogue; for example, ensuring consistency of generated responses with hand-engineered templates may help to improve a dialogue system's topical coherence (Grassi et al., 2021).

Seminal conversational systems such as ELIZA (Weizenbaum, 1966) operated on the basis of symbolic knowledge, allowing directly for persona development through the manipulation of ex-

plicit rules. However, this dependence on explicitly coded knowledge also rendered such systems knowledge-impoverished, and unable to make obvious inferences. Decades of AI research, aimed at solving this problem, have culminated in the creation of large language models (LLMs), and the emergence of in-context learning – i.e., steering the LLM towards particular behavior by including knowledge or examples in the natural language prompt (Brown et al., 2020). Recent work has found that systems that leverage in-context learning with LLMs outperform those that fine-tune smaller language models on conversational data (Madotto et al., 2021; Zheng and Huang, 2022).

While a variety of personas can be elicited from the information implicit in the weights of LLMs, the resulting personas are often unpredictable, opaque to dialogue designers, and prone to hallucination (Lim et al., 2022). Therefore, much recent work has focused on representing personas and knowledge explicitly, in a manner that can be leveraged by LLMs for generation using retrieval-in-the-loop methods (Shuster et al., 2021). Typically, these approaches represent personas using unstructured sets of natural language "facts" about an agent, possibly augmented with additional knowledge from a knowledge base.

In casual human-human dialogue, however, personas are often revealed through story-like narratives about experiences rather than one-off facts (Dunbar et al., 1997). For example, if a speaker mentions something involving sports, the interlocutor might respond by relating their typical experiences playing a sport in the past. These types of narratives, typically taking the form of "generic passages" (Carlson and Spejewski, 1997), often capture *habitual knowledge* – knowledge about the kinds of events that an agent participates in, or used to participate in. This knowledge includes the typical steps of a habitual event, as well as the typical goals, preconditions, and postconditions of the event. Originating from early research in artificial intelligence, *event schemas* have been proposed as a structured representation of the rich types of prototypical knowledge associated with generic and habitual events, such as causal and enabling relations, temporal relations, etc. (Chambers, 2013; Lawley et al., 2021; Li et al., 2021).

In this paper, we propose a novel approach to dialogue generation that uses a collection of explicit event schemas to augment an agent's persona, and that conditions an LLM to generate narrative-like responses consistent with these schemas through in-context prompting[1]. Furthermore, since it is often desirable for dialogue designers to be able to specify a persona using a small number of simple natural language facts, we propose a method for *bootstrapping* the creation of schemas from a set of simple facts. This method involves leveraging LLMs to first generate "generic passages" from the given facts, and then to induce structured habitual schemas from the passages – capturing both explicit steps from the passage and implicit knowledge associated with the event described by the passage. A high-level diagram of our approach is shown in Figure 1. We present evaluation results showing that the generated schemas are generally high quality, and can be used to condition LLMs to generate responses that are more diverse and engaging, yet also controllable.

## 2 Related Work

### 2.1 Persona-Based Dialogue Generation

Many past systems have attempted to integrate explicit customizable persona profiles with statistical response generation techniques; one of the earliest such systems was NPCEditor (Leuski and Traum, 2010), which used information retrieval (IR) to retrieve a hand-designed response from a persona, but was limited to question-answering dialogues. Attempts to make persona-based generation more general and robust were initially based on encoding personas as a single vector in sequence-to-sequence architectures (Li et al., 2016b; Kottur et al., 2017).

More recently, efforts have focused on making use of personas more directly: Zhang et al. (2018) crowd-sourced a large dataset of persona-based dialogues in which personas were represented as unstructured sets of natural language facts, and created a Seq2Seq model that uses IR to retrieve relevant persona facts as input. Subsequent studies built upon this approach using different models and extended persona datasets (Mazaré et al., 2018; Qian et al., 2018; Madotto et al., 2019; Zheng et al., 2019; Su et al., 2019; Salemi et al., 2023). However, while such approaches are effective at making responses conform to a particular persona, the generated responses are often shallow due to the simplicity of the persona representation.

---

[1]Code can be found at https://github.com/bkane2/habitual-response-generation

Some studies have shown that adopting richer persona representations that blend persona information with general world knowledge lead to more interesting and consistent responses (Majumder et al., 2020; Lim et al., 2022; Oh and Kim, 2022). Along these lines, Majumder et al. (2021) demonstrated that by sampling background stories relevant to retrieved persona facts from an external story corpus, a conversational model could generate responses that are more diverse and engaging than by using the persona alone. However, since the story corpus in this work was unrelated to the personas, there is a risk that a selected story may not fully cohere with the given persona. Moreover, using the story directly may leave out knowledge that is implicit but not necessarily expressed in the story, such as the underlying goals of participants. We build on this work by considering each story as a latent step in inducing a schema that contains both implicit and explicit knowledge associated with the story.

## 2.2 Deriving Symbolic Knowledge from Large Language Models

The framework of deriving explicit knowledge from LLMs has been explored in other work, though primarily in the context of IR and commonsense reasoning systems rather than persona-based dialogue generation. West et al. (2022) show that implicit knowledge within an LLM can be distilled into a symbolic commonsense knowledge graph using prompt engineering techniques. Other work focuses specifically on the problem of event schema induction using a neuro-symbolic pipeline (Lawley and Schubert, 2022) or zero-shot incremental prompting techniques (Dror et al., 2023; Sha, 2020). However, these studies focused on the induction of *generic* event knowledge (e.g., the steps typically taken to plan a wedding), rather than the *habitual* event knowledge implicit in a specific persona (e.g., a persona's typical experiences when attending weddings in the past).

## 3 Method

Given a dialogue context $\mathcal{U} = \{u_1, u_2, ..., u_{n-1}\}$ containing system and user utterances, our goal is to generate a response $u_n$ that utilizes knowledge from a relevant event schema $S^R \in \mathcal{S} = \{S_1, S_2, ..., S_m\}$ – this schema represents knowledge about a habitual activity that is part of the speaker's persona and that is relevant to the previous turn $u_{n-1}$. We ensure that the selected schema

is relevant using a multi-level information retrieval system to embed both the event schemas (treated as individual documents) and the knowledge contained within each event schema (treated as collections of documents), and to rank the schemas in $\mathcal{S}$ based on similarity to the embedding for $u_{n-1}$.

Following (Zheng and Huang, 2022), we employ a prompting-based approach in which a pre-trained LLM is used to produce a response utterance, provided a prompt that is dynamically constructed from the dialogue history and the selected schemas.

### 3.1 Schema Induction

Since structured event schemas for habitual activities are typically expensive for dialogue designers to create, requiring reasoning about causal relations and other implicit knowledge, we focus on the problem of automatically inducing event schemas from an unstructured persona $\mathcal{P} = \{p_1, p_2, ..., p_n\}$, where $p_i$ are natural language "facts" such as "I like to play tennis."[2]. Formally, we represent an event schema as a tuple $\langle \mathtt{H}, \mathtt{Pr}, \mathtt{S}, \mathtt{Po}, \mathtt{G}, \mathtt{E} \rangle$. $\mathtt{H}$ is a schema *header*; a sentence characterizing the overall schema event. $\mathtt{Pr}$, $\mathtt{S}$, and $\mathtt{Po}$ are sets containing schema *preconditions*, *static conditions* (conditions expected to hold throughout the overall event), and *postconditions*, respectively. $\mathtt{G}$ is a set containing typical goals of participants of the event, and $\mathtt{E}$ is a set containing typical episodes (i.e., substeps) of the event. We show an example of an event schema in Figure 2.

In order to generate sufficiently interesting and accurate schemas, we employ the method of *latent schema sampling (LSS)* introduced in (Lawley and Schubert, 2022) – this method regards an LLM, when conditioned on a schema header, as implicitly characterizing a distribution over stories sampled from that distribution. A full schema can then be induced from the sampled stories.

Thus, for each $p_i \in \mathcal{P}$, we sample $N_p$ stories (specifically, *generic passages* (Carlson and Spejewski, 1997) describing the typical process of a habitual event) using the GPT-3.5-TURBO LLM[3]. We use a few-shot prompt in which the LLM is supplied with a short definition of a generic passage, followed by $K_p$ examples. In contrast to the neuro-symbolic pipeline in (Lawley and Schubert,

---

[2]These facts may be hand-designed by a dialogue designer, crowdsourced (as in (Zhang et al., 2018)), or generated by an LLM.

[3]https://platform.openai.com/docs/models/overview

```
:header "I work in a bookstore."

:preconditions (
  "My shift has started."
)

:static-conditions (
  "The bookstore is stocked with books."
  "Customers visit the bookstore."
  "I am knowledgable about books and customer service."
)

:postconditions (
  "My shift at the bookstore is over."
  "Some customers have purchased books."
)

:goals (
  "My goal is to assist customers in finding the books they are looking for."
  "The customers' goal is to find the books they want to purchase."
)

:episodes (
  "Customers come looking for new titles to add to their collection, or to browse."
  "I welcome the customers and ask if they need any assistance."
  "I help the customers find books by using my knowledge of the store's inventory."
  [...]
  "I organize the bookshelves when the customers are not in the store."
)
```

Figure 2: An example of an event schema for a habitual "work at bookstore" activity. Note that some episodes are omitted for brevity.

2022), we leverage the in-context learning capabilities of GPT-3.5-TURBO to directly induce an event schema from a set of $N_p$ passages, given an abstract schema template and $K_s$ in-context examples[4]. See Appendix B.2 for our specific prompts, and Appendix C.2 for additional examples of generated schemas.

## 3.2 Dialogue Generation

We use the GPT-3.5-TURBO LLM to generate fluent responses, conditioned on a prompt containing a subset of the knowledge contained within a retrieved schema. Additionally, in order to allow controllable dialogue flow management – which is necessary for usability in many applied domains (Grassi et al., 2021) – we allow for two modes of generation: *unconstrained generation*, in which the LLM is prompted with the entire dialogue history and generates the next utterance without any constraints (apart from the retrieved knowledge); and *few-shot paraphrase generation*, where the LLM is prompted with a given sentence to paraphrase along with several in-context examples. In prac-

tice, the mode of generation may be mediated by a dialogue manager that manages the conversation flow and provides "raw" utterances (which may, for instance, be programmed by dialogue designers) to be selected for paraphrasing. For the purposes of this paper, we assume that, in the case of paraphrase generation, we have raw utterances available.

### 3.2.1 Schema Retrieval

As a first step in constructing a prompt, we use a multi-level retrieval system that uses a pretrained Sentence Transformer model[5] (Reimers and Gurevych, 2019) to embed and retrieve relevant schema knowledge. We pre-compute embeddings for each schema, as well as for each fact within each schema. For each dialogue turn, we also compute an embedding of the previous utterance $u_{n-1}$:

$$
\begin{aligned}
\boldsymbol{e}_{S_i} &= T(S_i) & \forall S_i \in \mathcal{S} \\
\boldsymbol{e}_{S_i, f_j} &= T(f_j) & \forall f_j \in S_i, \forall S_i \in \mathcal{S} \\
\boldsymbol{e}_{u_{n-1}} &= T(u_{n-1})
\end{aligned}
$$

---

[4]In practice, we found $N_p = 1$, $K_p = 2$ and $K_s = 1$ to be sufficient to produce accurate generations.

[5]https://huggingface.co/sentence-transformers/all-distilroberta-v1

Here $T$ is a Sentence Transformer model, $\mathcal{S}$ is the full set of schemas, $f_j \in S_i$ is the full set of facts contained within each section of schema $S_i$.

We then retrieve the single schema most relevant to $u_{n-1}$ using a cosine similarity measure, and score the facts within that schema based on computed similarity:

$$\text{score}(f_j) = \text{sim}(\boldsymbol{e}_{S^R, f_j}, \boldsymbol{e}_{u_{n-1}}) \qquad \forall f_j \in S^R$$
$$S^R = \underset{S_i \in \mathcal{S}}{\arg\max}\, \text{sim}(\boldsymbol{e}_{S_i}, \boldsymbol{e}_{u_{n-1}})$$
$$\text{sim}(\boldsymbol{e}_1, \boldsymbol{e}_2) = \frac{\boldsymbol{e}_1 \cdot \boldsymbol{e}_2}{\|\boldsymbol{e}_1\|\|\boldsymbol{e}_2\|}$$

The top $N_f$ facts according to score[6] are retrieved to be used in the prompt.

### 3.2.2 Unconstrained Generation

In the case of unconstrained generation, we sample a response from the LLM by prompting it with the full dialogue history, after conditioning on facts from the relevant habitual schema and the current dialogue schema:

$$u_n \sim \text{LLM}(F_R + F_D + \mathcal{U}),$$

where $F_R = \{f_1, ..., f_{N_f}\} \subset S^R$ are the relevant facts retrieved in the previous step, $F_D = S^D \setminus \text{E}(S^D)$ are all non-episodic facts from the current dialogue schema (i.e., preconditions, goals, etc.), and $\mathcal{U} = \{u_1, ..., u_{n-1}\}$ is the dialogue history.

### 3.2.3 Few-shot Paraphrase Generation

In the case of paraphrase generation, we employ a few-shot prompting strategy to condition the LLM to paraphrase the given sentence in a manner that is interesting, appropriate, and that makes use of the relevant facts. Specifically, in addition to the inputs used in the unconstrained setting, we format several in-context paraphrase examples along with a "raw" utterance to paraphrase, given the actual dialogue context:

$$u_n \sim \text{LLM}(F_R + F_D + \mathcal{E} + \mathcal{U} + \hat{u}_n),$$

where $\hat{u}_n$ is the sentence to paraphrase, and $\mathcal{E}$ is a set of $K_e$ in-context examples: $\mathcal{E} = \{(\mathcal{U}^1, \hat{u}_n^1, u_n^1), ..., (\mathcal{U}^{K_e}, \hat{u}_n^{K_e}, u_n^{K_e})\}$.

Examples of both types of prompts can be found in Appendix B.2.

---

[6]Excluding the schema header, which is always included in the prompt.

| | BASE | UNCS | PARA |
|---|---|---|---|
| Base Persona | ✓ | ✓ | ✓ |
| Dialogue History | ✓ | ✓ | ✓ |
| Event Schema | ✗ | ✓ | ✓ |
| Raw Response | ✗ | ✗ | ✓ |

Table 1: A summary of the differences in the resources available to each method that we compare in our evaluations. Note that each method in the order presented has access to all resources available to the previous method.

## 4 Experiments

We first evaluate our response generation method according to the following desiderata: (1) the generated responses improve diversity of output; (2) the generated responses are engaging, interesting, and relevant given the previous conversation, and (3) the generated responses are controllable; i.e., a dialogue designer can ensure that the responses still correctly express an intended response. The specific hyperparameter values that we use for our experiment are shown in Appendix A.

Since an important advantage of our approach is the reusability of the generated schemas for downstream tasks (e.g., for inferring additional facts from a dialogue agent's experiences), we also conduct an evaluation of the quality of the generated schemas − specifically, whether the facts within the schema correctly represent typical knowledge associated with the event that the schema describes.

### 4.0.1 Dataset

We conduct our experiment using the PersonaChat dialog dataset[7] (Zhang et al., 2018). We generate schemas and evaluate the performance of our response generation method using the test split, containing of 131,438 unique utterances. When evaluating our paraphrase generation method, we use the gold response annotations from the PersonaChat dataset for the raw utterances that are input to the model.

### 4.0.2 Baselines

We consider two baselines for evaluating the performance of our approach: First, we use the GPT-3.5-TURBO LLM without schema retrieval, provided only with the base persona and dialogue history in the prompt (BASE). The specific baseline prompt is shown in Appendix B.2. Second, we consider

---

[7]https://huggingface.co/datasets/bavard/personachat_truecased

the human-generated gold utterances from the PersonaChat dataset themselves (**GOLD**) as a baseline for our diversity, engagement, and relevancy metrics. Against these, we compare our two generation methods: unconstrained generation **UNCS** and paraphrase generation **PARA**. The differences between the three generation methods are summarized in Table 1 for reference.

## 4.1 Response Generation Evaluation

### 4.1.1 Automatic Evaluation

Following prior work (Majumder et al., 2021; Li et al., 2016a), we use several methods to measure the diversity of the generated outputs, per desideratum (1). First, we compute the mean percentage of unigrams and bigrams in the generated outputs that are distinct relative to the total number of generated words, reported as **D-1** and **D-2** respectively. We also report the mean lengths of the outputs as **Length**. Since the distinct n-gram measures do not represent the actual frequency distributions of words (and will tend to be penalized with longer responses), we also report the mean **ENTR** score across outputs – calculated as the geometric mean of entropy values of n-gram frequency distributions, for $n \in \{1, 2, 3\}$.

In order to test the controllability of our paraphrase generation method against other baselines, per desideratum (3), we also report several text similarity methods computed between a generated output and the gold PersonaChat response. We report widely-used n-gram-based similarity metrics such as **BLEU**, **ROUGE-L**, and **METEOR**, as well as the cosine similarity between contextualized embeddings produced by the ALL-DISTILROBERTA-V1 Sentence Transformer model (Reimers and Gurevych, 2019) (**ST**). However, since not all sentences in a generated response may be directly related to the gold response (e.g., an acceptable paraphrase may consist of a story followed by the intended response), it is difficult to interpret these metrics on the level of the full response. Hence, we compute the maximum *pairwise* similarity for each full sentence[8] between the generated and gold responses, and report the average value across all responses.

These results are shown in Table 3. We observe that the methods that use event schemas for generation generate responses with higher diversity than

---

[8]Split based on ".", "?", and "!" punctuation, filtering out sentences less than 5 words in length.

the baseline methods that do not have access to the schemas, as measured by D-2 and ENTR (although D-1 tends to favor the methods that generate responses that are shorter and therefore have a higher relative fraction of distinct uni-grams). Furthermore, we observe that the paraphrase generation method achieves considerably higher similarity to the gold responses than both the baseline and unconstrained methods (which perform comparably well on this metric).

### 4.1.2 Human Evaluation

To assess desideratum (2), we conduct a human evaluation of 100 randomly sampled examples on two metrics associated with response quality, following prior work (Majumder et al., 2021) – namely, whether the generated responses are **engaging** and **relevant** given the dialogue context. Annotators are tasked to make a pairwise comparison between responses from a pair of generation methods. We first collect annotations comparing the two baseline methods; under the assumption of transitive preferences, we then use the "winning" baseline as a comparison for each proposed method. We hired two Anglophone annotators for every sample; further details of our evaluation setup, including a screen capture of the task, are shown in Appendix B.

Our results are shown in Table 2, with starred values indicating differences that are significant with $p < 0.05$, using non-parametric bootstrap tests on 2000 subsets of size 50. The collected annotations are fairly noisy, with inter-annotator agreement (Krippendorff's alpha) being 0.21 and 0.23 for "engaging" and "relevant", respectively.

Despite this, we were able to observe moderate and statistically significant preferences for both the paraphrase and unconstrained methods over the LLM baseline in terms of engagement, and for the paraphrase method over the baseline in terms of relevancy. The LLM baseline itself was, in turn, significantly preferred over the gold responses for both questions. We believe that this can be attributed to the relatively short length and low diversity of language of the gold responses (as indicated in Table 3), as well as the ability of LLMs to interpolate smoothly with conversation history, even when constrained by our proposed methods.

We note, however, that many annotators were indifferent between the different generation methods. This is plausibly due to the fact that, generally, multiple response strategies are considered

| PARA vs. | UNCS | | BASE | | UNCS v BASE | | BASE v GOLD | |
|---|---|---|---|---|---|---|---|---|
| **Metric** | win | loss | win | loss | win | loss | win | loss |
| Engaging | 34.7 | 27.4 | 46.8* | 21.1 | 39.5* | 24.2 | 43.0* | 23.0 |
| Relevant | 33.2 | 23.7 | 44.7* | 24.2 | 37.4 | 22.6 | 40.5* | 25.0 |

Table 2: Pairwise comparisons between responses generated from each method (% win/loss, leaving ties out). Entries with * are statistically significant with $p < 0.05$ using a non-parametric bootstrap test on 2000 subsets of size 50. Additional details of the human evaluation are in Appendix B.

| Method | GOLD | BASE | UNCS | PARA |
|---|---|---|---|---|
| **Diversity** | | | | |
| LENGTH | 50.1 | 122 | 303 | 372 |
| D-1 | **97.0** | 93.8 | 81.7 | 78.9 |
| D-2 | 88.9 | 94.2 | 96.0 | **96.7** |
| ENTR | 2.20 | 2.91 | 3.61 | **3.84** |
| **Controllability** | | | | |
| BLEU | - | 1.25 | .843 | **8.60** |
| ROUGE-L | - | 19.3 | 19.8 | **34.6** |
| METEOR | - | 14.6 | 16.5 | **33.2** |
| ST | - | 35.6 | 35.0 | **55.6** |

Table 3: Diversity and controllability metrics on the PersonaChat test set. D-1/2 are the % of distinct uni- and bi-grams; ENTR is the geometric mean of n-gram entropy. BLEU, ROUGE-L, and METEOR are standard n-gram-based similarity metrics, and ST is the Sentence Transformer similarity measure. All similarities are calculated as average maximum pairwise values across sentences in each response. Best scores are bolded.

acceptable for the open-ended conversations in the PersonaChat dataset, and attests to the capability of LLMs to generate suitably engaging and relevant responses across prompting strategies.

## 4.2 Schema Evaluation

We evaluate the quality of the schemas, in themselves, through another human evaluation. We randomly select a subset of 200 individual schema facts from all generated schemas, each paired with the header of the schema it was taken from. An equal number of facts are selected for each type of schema relation. As a baseline, we select another 200 facts from the generated schemas, but randomly swap schema headers so that facts are paired with headers from unrelated schemas. For each type of schema relation, given a fact of that type and a schema header, we hire two Anglophone annotators to rate, on a 5-point Likert scale, how typical the fact is of an event described by the schema header. For instance, for a "static-condition" fact,

**Persona:**
I do not care about fashion.
I work in a bookstore.
...

**User:** I worked at Petsmart for two years and then I got canned.

**GOLD:** Ah too bad! I work at a barnes and noble. Chemistry grad.

**BASE:** Oh really? Why did you get fired?

**UNCS:** Oh no, sorry to hear that. But hey, maybe your knowledge of pets could come in handy if you're interested in finding some books on the subject. As someone who works at a bookstore, I'm pretty knowledgeable about different books [...]

**PARA:** That's tough. I actually work at a barnes and noble bookstore. Although, I didn't major in English or literature. I graduated from college with a chemistry degree. When I'm not shelving books, I help customers find what they're looking for [...]

Table 4: Sample generations by different methods, using the schema from Figure 2. Content in the gold utterance preserved by the paraphrase method are highlighted in red. See Appendix C.2 for additional examples.

an annotator might be asked "How typical is it that Sentence 2 is true throughout the duration of the event in Sentence 1?". Our full list of questions, and additional details of our evaluation setup, are shown in Appendix B.

The mean Likert ratings for the baseline and the generated schemas are shown in Table 5. All differences are significant with $p < 0.05$ using a Mann Whitney U test. We observe that the generated schemas are generally found to contain facts that are typical of the described event, relative to the randomized baseline. The smallest typicality

| Relation | BASE | SCHEMA |
|---|---|---|
| PRECONDITIONS | 2.51 | 3.65 |
| STATIC-CONDITIONS | 2.89 | 3.74 |
| POSTCONDITIONS | 2.93 | 3.23 |
| GOALS | 2.99 | 3.55 |
| EPISODES | 2.80 | 3.36 |
| **ALL** | 2.82 | 3.50 |

Table 5: Mean Likert ratings for the baseline and generated schemas; both the aggregate value and disaggregated values across schema relation types are shown. All differences are statistically significant with $p < 0.05$.

differences were observed for the "postcondition" relation, suggesting that inferences of this type may be more complex than other schema relations.

### 4.3 Qualitative Analysis

Table 4 shows generated responses from different methods for a particular persona and context. Qualitatively, we observe that the two models that are conditioned on the habitual schema from Figure 2 are able to generate longer and more detailed responses, making use of generic knowledge such as that people who work at bookstores can generally help customers find books of interest. On the other hand, the baseline model tends to generate responses that are fairly short and open-ended[9]. Furthermore, we observe that the paraphrase method is more frequently able to preserve the meaning of the intended raw utterance, as indicated.

### 5 Conclusion and Future Work

In this work, we demonstrated that habitual knowledge in the form of explicit event schema representations could be used to condition LLMs to generate more diverse and engaging dialogue responses. We experimented with two generation settings, one of which furthermore allows for a greater degree of controllability by a dialogue designer who may wish to provide intended utterances for the LLM to paraphrase. Moreover, to ease the burden of schema design, we proposed a novel method of inducing schemas from a base persona using an LLM through sampling "generic passages" about habitual activities.

Although the inclusion of habitual knowledge

---

[9]One important caveat is that this behavior is not necessarily undesirable; short open-ended questions can often be used in a conversation to demonstrate interest or empathy towards the interlocutor, although in this paper we are focused on the challenge of generating more engaging responses.

can be used to produce more engaging responses, it is not sufficient – often, conversations focus around more specific experiences and memories, and the knowledge captured by schemas generated with our approach can be somewhat banal. In future work, we aim to extend our approach to generate schemas that capture *atypical* aspects of an agent's experience with a particular kind of event, as well as more ordinary memories or knowledge. We also aim to incorporate this response generation mechanism into a broader dialogue management framework that allows for a higher-level decision about whether responding using a habitual schema is appropriate given a particular context.

### Limitations

We acknowledge several limitations of our proposed approach. First, given that our approach relies on the use of LLMs in both schema induction and response generation, it is limited by the inherent tendency of LLMs to hallucinate information (Ji et al., 2023). Although in our qualitative analysis we did not encounter many instances of the LLM fabricating wholly false information, this tendency presented itself in more subtle ways – particularly in the paraphrase model due to the complexity of the prompt. For example, if the sentence to paraphrase contains a first person pronoun, the LLM occasionally might reverse the pronoun in the generated response, falsely attributing some fact to the user instead. In some cases it may ignore the sentence to paraphrase altogether.

Second, although our method succeeds at generating more diverse, and engaging responses, this can often be inappropriate in certain conversational contexts, such as in a scenario that calls for a short affective response from the agent rather than a lengthy narrative-like response. Moreover, such responses may become repetitive over the course of a full conversation. Our approach would likely need to be integrated into a broader dialogue manager architecture in order to be usable in practice.

Third, the inference time and costs associated with LLMs (see Appendix B for estimates from our experiments) may make it difficult to use this approach at scale, or to generate schemas in an online manner.

Finally, we note that our experiments were limited to the English language; the performance of our approach may degrade if applied to lower-resource languages.

## Ethics Statement

Prior work has found that stories generated by LLMs can reinforce potentially harmful social biases (Lucy and Bamman, 2021). Since our work involves the use of LLMs to generate story-like passages as an intermediate step in deriving schemas, the resulting schemas would likely need to be carefully vetted to ensure that they do not contain harmful information. Furthermore, due to the possibility of hallucination described in Section 5, our approach should not be used in high-impact applications where failure to correctly paraphrase an intended response may incur heavy costs.

## Acknowledgments

This work was supported by DARPA under grant W911NF19-1-0029 and NSF Award IIS-1750380.

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

## A Implementation Details

### A.1 LLM Hyperparameters

We use the GPT-3.5-TURBO LLM for all generation, with 2048 max tokens. We use the default hyperparameters, i.e., a temperature of 1, top p 1, frequency penalty 0, and presence penalty 0. For the response generation prompts, we use stop sequences corresponding to the agent names.

### A.2 Experiment Hyperparameters

For schema induction, we use $k_p = 2$ examples for each passage generation prompt, and $k_s = 1$ examples for each schema induction prompt. We also set $N_p = 1$, i.e., we generate a single passage for each fact/schema. For generation, we use $k_e = 3$ examples for each paraphrase prompt, and retrieve $N_f = 5$ facts from the selected habitual schema (excluding the header). These values were found to be sufficient through preliminary sensitivity analysis.

## B Experiment Details

### B.1 Experiment costs

We estimate that generation of schemas for every item in the PersonaChat test set cost approximately $11, and took about 16 hours to complete (with OpenAI queries being sent in sequence from a singular process). Generating responses for all three methods for every item in the dataset cost approximately $8, and took about 14 hours to complete.

### B.2 Human Evaluation Setup

For our human evaluation of the generated responses, we used Amazon Mechanical Turk to hire two Anglophone (Lifetime HIT acceptance % > 98) annotators to rate batches of 5 pairwise comparison between generated responses. Our study participants were limited to native English speakers within the United States. Participants were compensated at a rate of $8.4 per hour for each assignment, and on average took about 1 minute to complete each assignment.

The comparisons were shuffled randomly between Human Intelligence Tasks (HITs), and the A/B responses were also swapped randomly. Figure 3 shows a sample HIT for comparison between two generated responses on engagement and sensibility.

For our schema evaluation, we ask annotators (using the same qualifications) to rate batches of 10

fact/header pairs, randomly shuffled between HITs. We asked the following questions for each relation type:

- **Preconditions** : "How typical is it that Sentence 2 is a pre-condition of the event in Sentence 1?"

- **Static-conditions** : "How typical is it that Sentence 2 is true throughout the duration of the event in Sentence 1?"

- **Postconditions** : "How likely is it that Sentence 2 is a result of the event in Sentence 1?"

- **Goals** : "How likely is it that Sentence 2 is a goal of the agent of the event in Sentence 1?"

- **Episodes** : "How likely is it that Sentence 2 occurs as a step of the event in Sentence 1?"

We used a 5-point Likert scale with the following labels:

1. very non-typical

2. somewhat non-typical

3. neutral

4. somewhat typical

5. very typical

Figure 4 shows a sample HIT for annotating a schema fact.

## C Language Model Prompts

We include examples of the prompts used in our method. For GPT-3.5-TURBO inputs, we use headers to distinguish inputs using system, user, and assistant roles. Variables to be filled in with specific content are shown using angle brackets.

### C.1 Schema Induction

In Figure 5 and Figure 6, we show prompts that are used to generate generic passages and to derive schemas from passages, respectively.

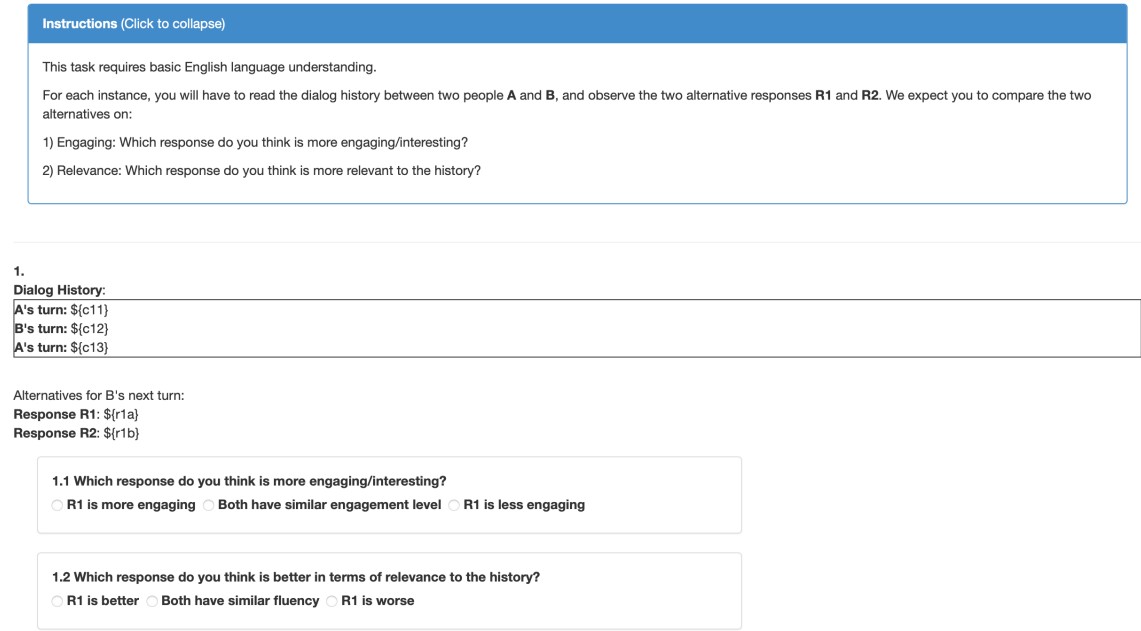

Figure 3: The human evaluation interface that we use for collecting pairwise comparisons between response generation methods. Variable $c_{ij}$ is replaced with the $j$th turn for item $i$, while $r_{ia}$ and $r_{ib}$ are replaced with the response candidates for item $i$.

## C.2 Dialogue Generation

In Figure 7 and Figure 8, we show prompts that are used to generate responses in the unconstrained generation mode and the few-shot paraphrasing mode, respectively. In both cases, `<background-user>` and `<background-sys>` are replaced with basic facts about the user and system (in our evaluation, we use the basic personas from PersonaChat[10], while `<habitual-facts>` is replaced with the selected facts from the retrieved habitual schema. The baseline prompt that we use in our evaluation is shown in Figure 9.

## D Examples

We present additional examples (hand-selected) of generated responses for each method on the PersonaChat dataset in Tables 6, 7, 8, 9, and 10. We show examples of both good responses, and responses reflecting common issues that we encountered (discussed in Section 5). We also show additional examples of generated schemas in Figures 10 and 11.

---

[10]In practice, these may also contain facts controlled by the dialogue manager, such as the current goals and common ground in the conversation.

**1.**
Sentence 1: ${s1_0}
Sentence 2: ${s2_0}

${qtext_0}
○ **Very non-typical**   ○ **Somewhat non-typical**   ○ **Neutral**   ○ **Somewhat typical**   ○ **Very typical**

Figure 4: The human evaluation interface that we use for collecting schema quality annotations. Variables $a1\_i$ and $a2\_i$ are replaced with the sentences for the schema header and fact for item $i$, respectively, while qtext$\_i$ is replaced with the corresponding question from the above list.

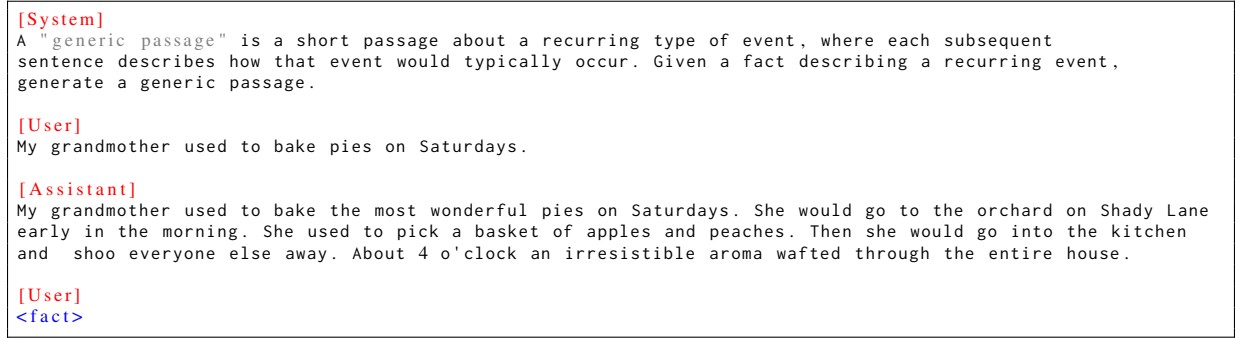

```
[System]
A "generic passage" is a short passage about a recurring type of event, where each subsequent
sentence describes how that event would typically occur. Given a fact describing a recurring event,
generate a generic passage.

[User]
My grandmother used to bake pies on Saturdays.

[Assistant]
My grandmother used to bake the most wonderful pies on Saturdays. She would go to the orchard on Shady Lane
early in the morning. She used to pick a basket of apples and peaches. Then she would go into the kitchen
and  shoo everyone else away. About 4 o'clock an irresistible aroma wafted through the entire house.

[User]
<fact>
```

Figure 5: The GPT-3.5-TURBO prompt that is used to generate a generic passage from a given persona fact.

```
[System]
Given a passage describing a habitual event, generate a schema consisting of the following sections:

:header
[predicate]([arguments])

:preconditions
[a list of preconditions of the event]

:static-conds
[a list of conditions that hold throughout the event]

:postconditions
[a list of conditions that hold after the event]

:goals
[a list of goals for all agents involved in the event]

:episodes
[a list of steps that occurred during this event]

[User]
My grandmother used to bake the most wonderful pies on Saturdays. She would go to the orchard on Shady Lane
early in the morning. She used to pick a basket of apples and peaches. Then she would go into the kitchen
and shoo everyone else away. About 4 o'clock an irresistible aroma wafted through the entire house.

[Assistant]
:header
– My grandmother used to bake pies on Saturday.

:preconditions
– It is Saturday
– My grandmother has collected the ingredients to bake a pie.

:static-conds
– My grandmother has the equipment to bake a pie.
– My grandmother knows how to bake pies.

:postconditions
– Pies have been baked.
– I have received pies.

:goals
– My grandmother's goal is to bake pies for me.
– My goal is to receive pies from my grandmother.

:episodes
– My grandmother goes to the orchard on Shady Lane.
– My grandmother picks a basket of apples and peaches.
[...]

[User]
<passage>
```

Figure 6: The GPT-3.5-TURBO prompt that is used to induce an event schema from a generic passage, using in-context examples. Note that some of the example schema is omitted for brevity.

```
[System]
Write a conversation between <user> and <sys>.

Background for <user>:
<background−user>

Background for <sys>:
<background−sys>

Use all of the following facts about <sys> in your response:
<habitual−facts>

<history>
```

Figure 7: The GPT-3.5-TURBO prompt that is used to generate a response in unconstrained generation mode, given names for the system and user, as well as the inputs described in Section 3.2.2.

```
[System]
<user> is having a conversation with <sys>.

Background for <user>:
<background-user>

Background for <sys>:
<background-sys>

Rewrite the sentences marked with [ORIGINAL] as [REWRITTEN]. Use a set of relevant facts in your rewritten
responses, but DO NOT change the meaning of the original sentence.

Relevant facts:
I enjoy skiing.
I went skiing in Utah last year.

Person B: What sorts of activities do you like?
Person A: I'm pretty into skiing.
Person B: Do you like video games?

<sys>: What sorts of activities do you like?
<user>: I'm pretty into skiing.
<sys>: That's awesome! i'm actually really into skiing as well. Just last year I went skiing in utah, which
        was a lot of fun. What are some other types of activities you like? do you like playing video games?

[...]

Relevant facts:
<habitual-facts>

<history>
Person B [ORIGINAL]: <sentence>

<history>
<sys> [REWRITTEN]:
```

Figure 8: The GPT-3.5-TURBO prompt that is used to generate a response in paraphrase generation mode, given names for the system and user, as well as the inputs described in Section 3.2.3.

```
[System]
Write a conversation between <user> and <sys>.

Background for <user>:
<background-user>

Background for <sys>:
<background-sys>

<history>
```

Figure 9: The GPT-3.5-TURBO prompt that is used as a baseline in the evaluation in Section 4.

```
:header "I bought my first home and I am unpacking and decorating."

:preconditions (
  "I have completed the process of buying a home."
  "My belongings have been moved into my new home."
)

:static-conditions (
  "I have unpacked boxes to unpack."
  "I have furniture and decor to set up."
)

:postconditions (
  "My belongings are unpacked and organized."
  "I can enjoy living in my new home."
)

:goals (
  "My goal is to unpack and organize my belongings."
  "My goal is to decorate and set up my home to my liking."
)

:episodes (
  "I take a deep breath and pick up a box."
  "I start looking for the perfect spot to set the box down."
  "I begin unpacking boxes and organizing my belongings."
  "I give extra attention to rooms like the kitchen and bathroom."
  "I start setting up furniture and decor."
  [...]
)
```

Figure 10: Example of a generated habitual schema, corresponding the example in Table 6.

```
:header "I like to take walks."

:preconditions (
  "It is early morning or late evening."
)

:static-conditions (
  "I have walking shoes."
  "I have access to different routes."
)

:postconditions (
  "I feel refreshed and clear-minded."
  "I have exercised."
)

:goals (
  "My goal is to clear my mind."
  "My goal is to receive exercise."
)

:episodes (
  "I put on my walking shoes."
  "I head out the door."
  "I walk for a few minutes, which are the toughest minutes of the walk."
  "I feel the cool breeze on my face and the warmth of the sun on my skin."
  "Sometimes I venture into the woods or through the park on my walks."
  [...]
)
```

Figure 11: Example of a generated habitual schema, corresponding to the example in Table 7.

**Persona:**
I am expecting twins in two months.
A already have a children.
I just bought my first home.
I work at a bank.
I am an omnivore.

**User:** Gosh. How exciting. And to think I was just looking forward to dancing tonight!

**GOLD:** Only dancing I do is down the hall to the bathroom in our new house.

**BASE:** Yes, it's definitely going to be an adventure! Enjoy your night of dancing though, sounds like fun.

**UNCS:** Sounds like a lot of fun! I'm actually busy unpacking and decorating my new home. Once everything is in place, I can finally relax and enjoy my new home. It's been quite the project to set up all of my furniture and decor, but now it's decorated and set up to my liking. How about you, do you have any exciting plans for the weekend?

**PARA:** That does sound like a fun time! The only dancing I do lately is down the hall to the bathroom in our new house. I just bought my first home and I'm in the process of unpacking and setting up furniture and decor. Once everything is in place, I can finally relax and enjoy my new home. It's definitely a busy time, especially with twins on the way, but I'm excited to have a space that is decorated and set up to my liking.

Table 6: Example generation where all methods yield reasonable responses.

**Persona:**
I like to eat hamburgers and french fries.
I don't drink soda.
I live in a house.
I like to take walks.

**User:** No, I'm single. I just graduated college. You?

**GOLD:** I am married and we just bought a house.

**BASE:** I'm a system, so I don't really have a personal life like humans do. But it's cool that you just graduated! What did you study?

**UNCS:** I am not capable of marriage, as I am a computer program. However, I do live in a house and I like to take walks. Sometimes I venture into the woods or through the park on my walks. It always leaves me feeling refreshed and clear-minded. I even have specific walking shoes for the job! Have you exercised today?

**PARA:** That's great to hear! I'm married and we actually just bought a house. Speaking of walking, I love taking walks myself. It's so refreshing and clears my mind. I make sure to wear my walking shoes, and sometimes I'll venture into the woods or through the park. Do you feel the same way after taking a walk?

Table 7: Example generation where the paraphrase method yields a more sensible response due to correctly paraphrasing the gold response.

**Persona:**
My pets name is charlie.
I produce music for artists.
My favourite food is pizza.
I drive a 2015 honda Civic.

**User:** I love winter thought, nice and cold.

**GOLD:** I do not enjoy the driving in my Honda civic.

**BASE:** I'm glad you enjoy it! I prefer warmer weather, but everyone has their own preferences. By the way, have you listened to any good music lately?

**UNCS:** That's great! Charlie loves winter too, that means more walks for him. He's my faithful companion and provides me with emotional support. I take care of his grooming so that he remains clean and healthy. Being with Charlie gives me joy and I'm grateful for the love and companionship he provides.

**PARA:** I actually don't enjoy driving in my Honda Civic, but one thing that always brightens up my day is taking care of Charlie, my pet. He provides me with so much love and companionship, and even emotional support when I need it. I make sure to groom his fur to keep him clean and take him out for walks to get some exercise. It's really rewarding to be a pet owner.

Table 8: Example generation where the unconstrained method yields a more sensible response, due to the paraphrase model missing the subtle implicature in the gold response (i.e., that the speaker doesn't enjoy driving due to the winter conditions).

**Persona:**
I love to go to Disney world every year.
Mickey mouse is my favorite character.
I play with my friends on the playground.
I'm in the third grade.

**User:** Hello, how are you doing today?

**GOLD:** Okay. I go to school. I like Disney world.

**BASE:** I'm doing great, thank you for asking! I always love talking to new people. How about you?

**UNCS:** Hello! I'm doing well, thank you for asking. I'm currently at school in the third grade, learning different subjects every day. Today I completed my spelling test and book report, and now I'm out for recess. It's a weekday, so I'll be returning to class soon to continue learning. How can I assist you?

**PARA:** I'm doing well, thank you for asking. Currently, I'm at school learning different subjects every day in the third grade. Today, I already completed my spelling test and book report. During recess, I like to go out and play with my friends on the playground. What about you? I know you love Disney world, is there anything else you enjoy doing?

Table 9: Example generation where the paraphrase method yields a poor response due to hallucination: the LLM attributes loving Disney world to the user, rather than to itself (per the gold response), even though this was never mentioned in the previous conversation.

**Persona:**
My favorite kind of movie is a documentary.
I'm pregnant with my first child.
I live with my husband in Pennsylvania.
I read a book every week.

**User:** That's exciting! Do you go to church?

**GOLD:** I do, me and my husband both go to Bible study every wednesday.

**BASE:** No, I don't go to church regularly. But I do believe in having faith and spirituality in my life. What about you?

**UNCS:** No, I am focusing on preparing for the baby's arrival and visiting the doctor for regular check-ups to ensure that my baby is healthy. I am also setting up a nursery and gathering necessary baby supplies. Additionally, I am staying active and taking care of my physical health so that I am prepared for motherhood. I cannot wait for the moment when I can hold my baby in my arms. I have been reading books and gathering information about motherhood to prepare myself as well.

**PARA:** Yes, I do go to church with my husband. I'm actually visiting the doctor regularly to make sure that my baby is healthy and preparing for the baby's arrival by setting up a nursery and gathering necessary baby supplies. I'm also staying active and taking care of my physical health. I can't wait until the moment when I can hold my baby in my arms! I've also been reading books and gathering information about motherhood.

Table 10: Example generation where both schema-based methods yield poor responses due to the retrieved schema not being relevant (although the paraphrase method is still able to convey the gold response).