# OpenReview forum: "We Are What We Repeatedly Do: Inducing and Deploying Habitual Schemas in Persona-Based Responses"
_EMNLP/2023/Conference — EMNLP 2023 Main_

### Official Review · Reviewer_qnGv · 2023-08-04

**Soundness:** 4

**Excitement:**

4: Strong: This paper deepens the understanding of some phenomenon or lowers the barriers to an existing research direction.

**Paper Topic And Main Contributions:**

The paper proposes a novel approach to dialogue generation for virtual conversational agents. The goal is to endow these agents with a specific persona and generate responses that are both natural and consistent with this persona. The approach leverages event schemas, representing habitual knowledge in narrative-like form, to augment the agent's persona. The authors introduce a method to bootstrap the creation of schemas from simple natural language facts. Through in-context prompting with large language models, the proposed approach generates more engaging and controllable responses, enhancing the performance of conversational agents. The main contributions include the use of event schemas to represent personas and knowledge explicitly, the bootstrapping method for creating schemas, and the improved performance of dialogue generation demonstrated through evaluation results.

**Questions For The Authors:**

A: In l. 194, the paper mentions the use of dialogue context – Do you only observe response utterances of the system or from the user as well, as it is not clear in the paper?
B: It is correctly pointed out that the approach leads to very long responses which may be inappropriate in certain situations. How do you plan to tackle this issue?
C: How could you mitigate the issues by including a gold standard for response evaluation?
D: How were the study participants compensated and could give more information about the demographics?

**Reasons To Accept:**

- Very interesting approach for generating more sophisticated personas for LLMs in a structured manner while relying on a strong theoretical background, i.e. the inclusion of habitual knowledge. Particularly, the use of event schemes seems to be a promising approach for generating more natural responses and making a system's persona more controllable
- Good reproducibility
- Rich evaluation including automatic and human evaluation
- Results suggest that the approach creates more engaging and diverse responses-

**Reasons To Reject:**

- At times lack of clarity:
            Introduction: The paper mentions that there is a complementary need to control the follow of dialogue to ensure topically coherent
            responses (l. 44 - 48), but then the topic is more or less dropped throughout the paper
            Methods: Here, there are some unclarities due to the many concepts mentioned in this section. Better illustrated examples could
            help to better understand the response generation process during dialogue.

- Inclusion of the gold standard in response generation seems to be a bit unfair, especially when measuring BLEU score, etc., as the responses then often contained the gold sentence, which naturally results in more similarity to the gold standard.

**Reproducibility:**

5: Could easily reproduce the results.

**Reviewer Confidence:**

4: Quite sure. I tried to check the important points carefully. It's unlikely, though conceivable, that I missed something that should affect my ratings.

---

> ### Author Rebuttal · Authors · 2023-08-28
>
> Thank you for the valuable feedback! We have refined the paper to address some of the questions and concerns raised by your review. We provide responses below.
>
> ---
>
> **1. Mentioning topical coherence in the introduction and dropping it in the remainder of the paper**
>
> Our intention with mentioning topical coherence in the introduction was primarily to motivate the need for allowing controllable dialogue flow, with the problem of topical coherence being given as a specific example where controllable dialogue flow may be desirable. We refer in our introduction to (Grassi et al., 2021) who, for instance, use an ontology of hand-engineered response templates to improve the topical coherence of their dialogue system. This need forms the motivation for our proposed paraphrase method, where a hand-engineered raw response is provided by a dialogue designer to be paraphrased into a more natural persona-based response.
>
> In order to clarify this, we have amended lines 44-48 to the following:
> *"Furthermore, in many practical applications beyond chit-chat, there is a complementary need to control the flow of dialogue; for example, ensuring consistency of generated responses with hand-engineered templates may help to improve a dialogue system's topical coherence (Grassi et al., 2021)."*
>
> **2. Better illustrated examples in the Methods section**
>
> If space in the camera-ready version allows, we can create an additional figure showing the architecture of our two response generation methods, using the same example from Figure 2 and Table 4.
>
> **3. Fairness of including the gold standard in response generation**
>
> We believe that there has been some confusion over the purpose of our inclusion of the gold standard in our paraphrase method, as a result of unclear language used in our paper.
>
> As mentioned in our response to the first point, the motivation for our paraphrase method is to allow conformity with an intended raw response, which may for instance be supplied by a dialogue designer. For our PersonaChat experiment, we make the assumption that we can regard the gold annotations from the dataset as raw responses to paraphrase -- in this sense, we do not actually view them as a true gold standard for responses (indeed, in our human evaluation, annotators tended to prefer even the LLM baseline to the gold annotations), but rather as purely an input to our paraphrase method.
>
> The purpose of our similarity-based evaluation metrics, then, is not to measure the overall quality of the generated responses (we instead do this through a combination of automatic diversity-based metrics and human judgements of engagement/relevancy), but rather to investigate whether the paraphrase method is indeed able to conform to the given raw response. To test this hypothesis, we measure similarity between the raw response and the generated response for the paraphrase method in comparison to the other methods. Since it is an open question whether an LLM is in fact able to maintain consistency with the raw response, especially when conditioning on additional habitual knowledge and conversation history, we believe that our results on these metrics present an interesting finding.
>
> We appreciate your feedback, as it prompted us to modify the paper to clarify the language used around the paraphrase method. Particularly, we have replaced mentions of "gold responses" with the more generic term "raw responses" throughout the paper, except for Section 4, where we have added an explicit statement that we use the gold annotations from PersonaChat as raw responses in our experiments.
>
> **4. Clarification questions**
>
> *A) Dialogue context*
>
> Our dialogue context contains both utterances from the system as well as utterances from the user (we have added this clarification to the paper as well).
>
> *B) The issue of lengthy responses being inappropriate in particular situations*
>
> While addressing this issue largely lies outside of the scope of our paper, we believe that our proposed response generation method should be situated within a dialogue management system that, at a minimum, should maintain some representation of the dialogue state, update the dialogue state based on interpretations of user input, and map the current dialogue state to suitable actions (cf. the minimal framework for dialogue management provided by [1]).
>
> Within such a dialogue system, the method used for generation may be determined by the dialogue manager based on relevant features of the dialogue and domain. A simple approach, for instance, would be using a classifier to select between our persona-based response generation methods or an alternative response generation method that produces e.g. short empathetic responses, open-ended questions, etc.
>
> [1] David R. Traum and Staffan Larsson. “The Information State Approach to Dialogue Management.” In: Current and New Directions in Discourse and Dialogue. Ed. by Jan van Kuppevelt and Ronnie W. Smith. Dordrecht: Springer Netherlands, 2003, pp. 325–353.
>
> *C) Mitigating the issues by including a gold standard for response evaluation*
>
> We hope that this has been adequately answered by our response to (3), but if not, please let us know and we will attempt to clarify further.
>
> *D) Compensation and demographics of participants*
>
> Our study participants were recruited using Amazon Mechanical Turk, and were limited to native English speakers within the United States. Participants were compensated at a rate of $8.4 per hour for each assignment, and on average took about 1 minute to complete each assignment. We have added the missing information to Appendix B.
>
> We hope this clarifies the reviewer’s concerns. Please don’t hesitate to let us know if you have any further questions.

---

### Official Review · Reviewer_gBoD · 2023-08-05

**Soundness:** 4

**Excitement:**

4: Strong: This paper deepens the understanding of some phenomenon or lowers the barriers to an existing research direction.

**Paper Topic And Main Contributions:**

The paper presents an approach to dialogue generation that uses a collection of explicit event schemas to augment an agent's persona and conditions an LLM to generate narrative-like responses consistent with these schemas through in-context prompting.

**Questions For The Authors:**

1-	How many facts do we need when generating new schemas?

**Reasons To Accept:**

1-	The paper is well structured. It presents clearly the main contributions;
2-	The method is well formulated;
3-	The paper presents several rich appendices that detail the method and experiments;


**Reasons To Reject:**

1-	Some passages of the paper should be clarified, such as schema generation and evaluation.

**Reproducibility:**

3: Could reproduce the results with some difficulty. The settings of parameters are underspecified or subjectively determined; the training/evaluation data are not widely available.

**Reviewer Confidence:**

3: Pretty sure, but there's a chance I missed something. Although I have a good feel for this area in general, I did not carefully check the paper's details, e.g., the math, experimental design, or novelty.

---

> ### Author Rebuttal · Authors · 2023-08-27
>
> Thank you for the valuable feedback! We have refined the paper to address the questions raised by your review. We provide responses below.
>
> ---
>
> **1. Number of facts needed when generating new schemas**
>
> We assume that each fact within a persona corresponds to exactly one schema, containing that fact as a header (i.e., the sentence characterizing the overall event; see lines 223-225). For example, the persona fact "I work in a bookstore." would be used to generate the schema shown in Figure 2. Since our method of schema induction involves first sampling passages from an LLM, each generated schema requires only the single fact that corresponds to the schema header. We have added this clarification to Section 3.1 of the paper.
>
> **2. Clarification of other details needed for reproducibility**
>
> We have clarified some points in the schema generation and evaluation sections with the help of the feedback in this review as well as the others; we refer to our rebuttals to the other reviews for additional details. In addition, we want to note that the parameters and data used in our experiment are detailed in the Appendices.
>
> If there are any other clarifications or details that we can provide to improve reproducibility of our work, please do not hesitate to follow up with further questions.

---

### Official Review · Reviewer_x1KL · 2023-08-06

**Soundness:** 3

**Excitement:**

4: Strong: This paper deepens the understanding of some phenomenon or lowers the barriers to an existing research direction.

**Paper Topic And Main Contributions:**

The presented work is based on the assumption that personas in realistic human dialogue are related to story-like narratives and involve rich habitual knowledge which can be induced from LLMs. These event schemas can then be used in dialogue generation by first calculating the cosine similarity between sentence embeddings of the previous utterance and the schema embeddings, and then by prompting a LLM (GPT-3.5 turbo) with the dialogue history, habitual facts from the previous dialogue step and the non-episodic facts from the current dialogue schema (unconstrained generation) as well as instructing the LLM to paraphrase a given sentence in a way that it is interesting and appropriate and uses relevant facts  (few-shot paraphrase generation). The approach is tested using the PersonaChat dialogue dataset. The evaluation experiments (using automatic methods and human evaluation) provide some evidence that generated schemas comprise facts that are considered more typical for the schema than a random combination of schema headers and facts. There is also some evidence that the use of event schemata in LLM prompting has positive effect on the diversity of generated dialogue utterances, and that utterances generated by the paraphrase generation method show higher similarity to human generated (gold) utterances from the PersonChat dataset than utterances generated by the other prompting methods.

**Reasons To Accept:**

Plausible approach to leverage persona-driven, event-related habitual knowledge in prompting LLMs for turn-level dialogue generation.  Well thought out experimental set up. Clear and concise presentation of the work done.

**Reasons To Reject:**

Effects are shown for English only. The experimentation results are ambiguous, and indicate the need for further investigation of LLM-based dialogue generation.

**Reproducibility:**

4: Could mostly reproduce the results, but there may be some variation because of sample variance or minor variations in their interpretation of the protocol or method.

**Reviewer Confidence:**

4: Quite sure. I tried to check the important points carefully. It's unlikely, though conceivable, that I missed something that should affect my ratings.

**Typos Grammar Style And Presentation Improvements:**

Line 277, 325, 365: briefly say where the respective gold utterances come from; unclear whether “”old utterance” always refers to utterances from the PersonaChat dataset.
Table 2: Any assumption why the gold response scores comparatively badly?
Table 5 Likert ratings: add information on the scale and wordings you used, e.g. not very typical (5), typical (4), neutral (3) …

---

> ### Author Rebuttal · Authors · 2023-08-27
>
> Thank you for the valuable feedback! We have refined the paper to address some of the presentation improvements suggested by your review. We provide responses below.
>
> ---
>
> **1. Source of gold utterances**
>
> In general, the "gold utterances" required for the paraphrase method may be supplied by an expert dialogue designer or crowdsourced within a particular domain (see lines 273-274). In our experiment, we make use of the existing gold annotations within the PersonaChat dataset.
>
> In order to clarify the language around the paraphrase method in our paper, we have replaced mentions of "gold utterances" with the more generic term "raw utterances" throughout Section 3 -- making clear that such raw utterances must be supplied as domain-specific inputs to our paraphrase method -- and have added a clearer explanation of how we derive the raw utterances from the PersonaChat gold utterances for our experiments in Section 4.
>
> **2. Interpretation of comparatively poor human evaluations of gold responses in Table 2**
>
> We believe that the crowdsourced PersonaChat gold responses were judged as less engaging than the LLM-based methods primarily due to their rather short length and relatively low diversity of language, as indicated by the metrics in Table 3. Furthermore, in a qualitative analysis of the gold responses, we observed that many were not fluent with the conversation history, but instead abruptly shifted the topic to one of the facts provided to the crowdsourced workers. Our assumption is that the LLM-based methods were judged as more relevant than the gold responses due to the ability of LLMs to interpolate smoothly with conversation history, even when constrained by our proposed methods. We have extended our discussion of the results in Table 2 in the paper.
>
> **3. Information on scale and wording for Table 5 Likert ratings**
>
> We have added the scale and wordings used for our Likert ratings to the paper, reproduced below:
>
> 1: very non-typical
>
> 2: somewhat non-typical
>
> 3: neutral
>
> 4: somewhat typical
>
> 5: very typical
>
> **4. Effects limited to English language**
>
> The authors of this paper have previously focused primarily on English language dialogue systems for several application domains, and our current contribution is largely motivated by an effort to improve the performance of these systems. However, we agree that further investigation of LLM-based dialogue generation across languages remains an important direction for future work, as noted in our discussion of limitations.
>
> We hope this clarifies the reviewer’s concerns. Please don’t hesitate to let us know if you have any further questions.

---

### Meta-Review · Area_Chair_iXYM · 2023-09-19

**Recommendation:** 4

**Metareview:**

The authors present a method to enable more consistent persona based dialogue generation for LLMs by incorporating situated, habitual knowledge in the form of schemas.

Most reviewers (at least the ones who have acknowledged the rebuttal) agree that this is a exciting approach to enable more complex personas in dialogue with LLMs, though there are some questions (partially answered) about human evaluation protocols and inclusion of gold responses, etc. especially as GPT 3.5 is used making the results somewhat irreproducable. Overall, I believe the pros of this approach outweighs the cons and merits presentation for further study.

---

### Decision · Program_Chairs · 2023-10-07

**Decision:**

Accept-Main

**Comment:**

The authors present a method to enable more consistent persona based dialogue generation for LLMs by incorporating situated, habitual knowledge in the form of schemas.

Most reviewers (at least the ones who have acknowledged the rebuttal) agree that this is a exciting approach to enable more complex personas in dialogue with LLMs, though there are some questions (partially answered) about human evaluation protocols and inclusion of gold responses, etc. especially as GPT 3.5 is used making the results somewhat irreproducable. Overall, I believe the pros of this approach outweighs the cons and merits presentation for further study.